# An Improved Perceptual Hash Algorithm Based on U-Net for the Authentication of High-Resolution Remote Sensing Image

**Kaimeng Ding [1,2,](#) , Zedong Yang [3,](#) , Yingying Wang [1] and Yueming Liu [2]**

[1]    Jinling Institute of Technology, Nanjing 211169, China
[2]    State Key Laboratory of Resource and Environment Information System, Institute of Geographic Sciences and Natural Resources Research, Chinese Academy of Science, Beijing 100101, China
[3]    Chinese Academy of Surveying and Mapping, Beijing 100039, China
*    Correspondence: dkm@jit.edu.cn (K.D.); yaogandasai@163.com (Z.Y.);
     Tel.: +86-181-6809-2159 (K.D.); +86-177-7783-1767 (Z.Y.)

**Abstract:** Data security technology is of great significance for the effective use of high-resolution remote sensing (HRRS) images in GIS field. Integrity authentication technology is an important technology to ensure the security of HRRS images. Traditional authentication technologies perform binary level authentication of the data and cannot meet the authentication requirements for HRRS images, while perceptual hashing can achieve perceptual content-based authentication. Compared with traditional algorithms, the existing edge-feature-based perceptual hash algorithms have already achieved high tampering authentication accuracy for the authentication of HRRS images. However, because of the traditional feature extraction methods they adopt, they lack autonomous learning ability, and their robustness still exists and needs to be improved. In this paper, we propose an improved perceptual hash scheme based on deep learning (DL) for the authentication of HRRS images. The proposed method consists of a modified U-net model to extract robust feature and a principal component analysis (PCA)-based encoder to generate perceptual hash values for HRRS images. In the training stage, a training sample generation method combining artificial processing and Canny operator is proposed to generate robust edge features samples. Moreover, to improve the performance of the network, exponential linear unit (ELU) and batch normalization (BN) are applied to extract more robust and accurate edge feature. The experiments have shown that the proposed algorithm has almost 100% robustness to format conversion between TIFF and BMP, LSB watermark embedding and lossless compression. Compared with the existing algorithms, the robustness of the proposed algorithm to lossy compression has been improved, with an average increase of 10%. What is more, the algorithm has good sensitivity to detect local subtle tampering to meet the high-accuracy requirements of authentication for HRRS images.

**Keywords:** high-resolution remote sensing image; deep learning; perceptual hash; integrity authentication; U-net

## 1. Introduction

As an important carrier of geospatial information, high-resolution remote sensing (HRRS) image has been widely used in many geoscience applications, including disaster assessments, mapping surveys, high-accuracy navigation, reconnaissance, monitoring, etc. Meanwhile, the development of geoscience information systems (GIS) and network technologies has provided advanced technical support for the sharing and using of HRRS images, but it also poses new challenges to the security of HRRS images, how to ensure the integrity of HRRS images is one of the key issues. HRRS images

generally have characteristics of high precision and confidentiality. If the HRRS images are tampered or their content integrity is questioned by users, their value of use will be greatly reduced. Examples of the tampering instances are shown in Figure 1. Even with careful inspection, it is difficult to confirm whether the HRRS images in Figure 1 have been tampered. In more serious cases, it will even affect national security.

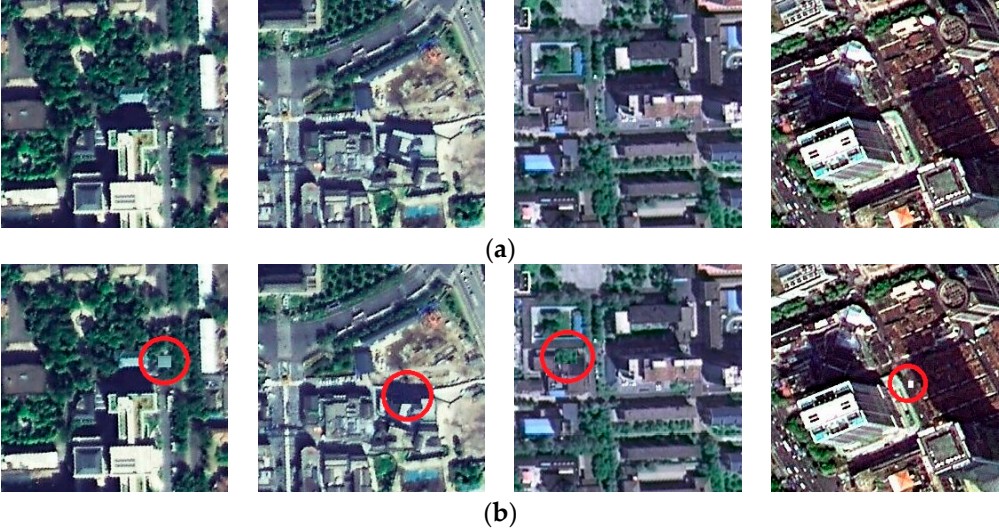

(a)

(b)

**Figure 1.** Examples of tampered HRRS images that have undergone different tampering techniques: (**a**) Original HRRS images, (**b**) corresponding tampered images.

In this case, authentication technology of HRRS images is becoming increasingly challenging. In fact, confidentiality and authentication are both aspects of information security, and they have different attributes. Confidentiality is to ensure that information is not stolen or illegally used. Authentication technology protects data by verifying the data integrity, the reliability, and authenticity of the data source.

As HRRS images and ordinary images are similar in existence, the image authentication methods can be used to achieve authentication of HRRS images. At present, image authentication methods mainly include cryptography signatures and watermarking techniques. Cryptographic hash functions and digital signature techniques are too sensitive to changes in the binary level of image data. As long as the data is altered by one bit, the data is considered to have been tampered with. This sensitivity is very suitable for the authentication of textual data, but not suitable for HRRS image authentication. In the actual use and transmission process of HRRS images, after the HRRS images are subjected to operations such as format conversion, data compression, or invisible watermark embedding, the useful content information carried by the HRRS images is not changed, and only the information carrier is changed. Watermarking technology (mainly fragile watermark) embeds the authentication information into an image, and when the image content is suspected, extracts the embedded authentication information to verify the integrity of the image. However, fragile watermark takes advantage of the nature of the watermark itself and cannot achieve content-based authentication for the image. What is more, fragile watermark will modify the original data, which is not allowed in many application environments for HRRS images.

In the field of GIS application, the authentication of HRRS images is essentially concerned with the integrity of the content, rather than the content carrier itself. Perceptual hashing provides a viable approach to content-based authentication for HRRS images. The perceptual hash algorithm, also known as perceptual hashing, is a one-way mapping of multimedia data to perceptual digests and can map the multimedia data with the same perceptual content to the same digital digest and satisfy perceptual robustness and security [1,2]. It has attracted increasing popularity in recent years [3–8].

HRRS images have the following characteristics, such as rich geometry, structure, texture features, fine-grained spectra, multi-scale target objects, etc., which makes the perceptual hash algorithm for HRRS images challenged.

In the related research for HRRS images, a Gabor filter bank and DWT-based perceptual hash algorithm is proposed in [9] for remote sensing image, while it is not specifically designed for HRRS images. In [10], a multiscale edge-feature-based perceptual hashing scheme is proposed for HRRS image, which has good sensitivity to detect subtle illegal tampering, while it adopts the traditional feature extraction method and cannot distinguish false edge information.

In the last few years, the deep learning represented by end-to-end learning of feature representation has achieved well performance in several image hash applications [11–16], but from the perspective of authentication for HRRS images, many challenges remain.

(1)    One of the most crucial challenges for applying deep learning to perceptual hash is that it requires a large number of training samples, otherwise it is easy to cause overfitting. However, there are only a few samples are available for each type of HRRS images to meet the authentication accuracy.

(2)    Existing deep-learning-based hash algorithms mainly focusing on the field of image retrieval [11–16], while image retrieval and image authentication are two different research areas, and the requirements of them are very different.

(3)    Interpretability has been identified as a potential weakness of deep neural networks [17], while it is necessary to illustrate the tampering of the images during the authentication process. Given the complexity, modern Earth system models are often also not easily traceable back to their assumptions in practice [18], which limits the interpretability of remote sensing image authentication.

(4)    The accuracy of the label of the training sample has a significant effect on the accuracy [19], while most of the existing research adopts the method of manually labeling samples, making the accuracy of sample information have a negative impact on image authentication.

In this paper, we proposed an improved perceptual hash scheme based on deep learning for the authentication of HRRS images by making use of the advantages of U-net model. U-net [20], which is an improved fully convolutional neural network (FCN) model, has a strong capability to automatically learn the high-level representations for images. We use U-net to extract the robust edge features of the HRRS image. The contributions of our work were elaborated as follows:

(1)    To the best of our knowledge, this is the first attempt in the GIS field to explore the application of U-net to the integrity authentication of HRRS images.

(2)    As high-precision training sample is an important guarantee for ensuring the accuracy of the authentication algorithm for HRRS images, a training sample generation method combining Canny operator and artificial processing was proposed.

(3)    A modified U-net model is studied to extract robust edge feature of HRRS images, which is the key step of perceptual hash.

The rest of this paper is organized as follows: Section 2 outlines the progress of related research. Section 3 presents our proposed method. Section 4 analyzes and discusses the experiment result. A discussion and a conclusion of our study are presented respectively in Sections 5 and 6.

## 2. Related Work

### 2.1. Perceptual Hash and HRRS Image

Tamper sensitivity and robustness are the two most important attributes of perceptual hash, but the two are often contradictory. Compared with cryptographic hash function, the most prominent feature of perceptual hash is that it has perceptual robustness, that is, after the image is subjected to the content-preserving operations (such as format conversion, watermark embedding, etc.),

the corresponding perceptual hash value does not change significantly. Similar to cryptographic hash function, perceptual hashing can map multimedia data (such as image and video) into a sequence of digests of a certain length, that is, the content information of the multimedia data can be represented with as little information as possible. What is more, perceptual hash is sensitive to the changes or tampering of the multimedia data and can realize integrity authentication based on perceptual content. According to different multimedia data, perceptual hashing mainly includes image perceptual hashing, audio perceptual hashing, and video perceptual hashing. Perceptual hashing can play an effective role in many areas, such as image retrieval, image authentication, copy detection, etc.

As the resolution of HRRS images continues to improve, the increasing complexity of HRRS images application and sharing environment, the integrity authentication of HRRS images are becoming more and more serious, and higher requirements are imposed on the perceptual hash authentication algorithm. Research on perceptual hash authentication algorithm focuses on the extraction method of image robust features and the effective dimensionality reduction coding method. In the research of perceptual hash for HRRS images, the key and difficult point is also how to extract the perceptual features that satisfy the integrity authentication requirements and generate a perceptual hash value that meets the actual needs. The existing perceptual hash algorithms based on the traditional feature extraction method still have some shortcomings in terms of robustness. For example, "false features" such as "noise" cannot be effectively identified. It is not possible to carry out targeted authentication for specific application environments.

The most prominent feature of HRRS images is the high spatial resolution and strong discrimination of ground objects. High spatial resolution is a relative concept that will constantly change standards as technology evolves. At present, the highest spatial resolution of the world commercial remote sensing satellite image has reached the sub-meter level: The resolution of the international commercial remote sensing satellite GeoEye has reached 0.41 m, and the WorldView-3 satellite has a resolution of 0.3 m resolution. Compared with low- or moderate-resolution-resolution remote sensing images, HRRS images display more detailed information.

The existing image perceptual hash algorithms do not take the data characteristics and application environment of HRRS images into account and cannot fully satisfy the perceptual-content-based authentication of HRRS images. As an important geospatial data, the high-resolution image has the following characteristics: High-resolution images have higher requirements for measurement accuracy, and image perceptual hash algorithms cannot meet the requirements of high-resolution images for measurement accuracy. The interpretation of HRRS image content has diversity and ambiguity, which increases the difficulty of the perceptual hash algorithm to extract the content features of HRRS images. Unlike medical images and ordinary optical images, HRRS images (especially the HRRS images with a large coverage) often have no unique subject information, which increases the uncertainty of the extraction of perceptual features of HRRS images. HRRS images also have other features such as scale and temporal, while ordinary images are not very obvious in this respect. In addition, HRRS images often have massive data characteristics, so they have high requirements for the "summary" of authentication information.

On the other hand, edge feature is capable of characterizing the integrity of the effective content of the HRRS images with high precision, and edge-feature-based perceptual hash algorithms can meet the high accuracy measurement requirement of HRRS image. What is more, edge feature plays an important role in the application of HRRS images, for example, edge feature is closely related to the object segmentation. If we detect a large change in the edge features of HRRS image, we can identify that the available content on the image has been tampered with. Compared with other kinds of perceptual hash algorithms, edge-feature-based perceptual hash algorithm can perform integrity authentication with higher precision.

However, the traditional edge detection approaches mainly use the differential operator such as Canny operator to perform edge detection. These detection operators extract the edge by detecting the sharp change of the local color and brightness of the image, which is easily affected by noise and will

produce false edges. It then affects the robustness of the perceptual hash algorithm. What is more, traditional edge detection approaches using pixel-level features have difficulties to distinguish the faint edge feature from the noise, and they not easy to represent object-level information [21], which make them not very directly usable for perceptual hash algorithms. Fortunately, the emerging deep learning (DL) theory provides a new and feasible solution to the above problems. Deep learning has strong capability to learn high-level representations of natural images automatically and there is a recent trend of using CNN to perform edge detection [21], which provide the possibility to extract the robust edge features of HRRS images for our work. Instead of the traditional methods, we use deep-learning-based method to extract robust edge features of the HRRS image.

## 2.2. Deep Learning

Deep learning is a form of machine learning (ML) approaches using artificial neural network (ANN) to learn hierarchical feature representations by building high-level features from raw data [18]. Deep learning has been successfully applied in many different fields [22–27]. The success of deep learning lies with access to large training datasets, which often consist of thousands, if not millions, of training examples [27], and for deep learning, the term "deep" refers to hundreds of layers.

### 2.2.1. Convolutional Neural Networks

Convolutional neural networks (CNN) is one of the most popular deep neural networks. To form specific network architectures, CNN consist of a series of layers like building blocks. It has started a deep revolution in deep learning given their capacity for processing image-like data using local connections, pooling, shared weights, and several layers [28]. It has been implemented as the base for various tasks such as object detection [29], video classification [30], super-resolution [31], and so forth.

Because of its low computational efficiency, fully convolutional networks (FCNs) [32] have recently has received extensive attention. FCN replaces the fully connected layer with convolutional layers to form a full convolutional network. To generate pixel-to-pixel translations between input and output images, FCN utilizes subsampling, up-sampling operations and sequential convolutional.

### 2.2.2. U-Net

Our approach for extracting edge features uses an improved FCN model, U-net [20] model. U-net model consists of symmetric contracting path and expansive path, so it is characterized by the U-shaped architecture.

U-net model consists of a series of kernels acting as filters to map out particular features of the input image. It has two main components [33]: A "down" component and an "up" component. The former one is applied to detect desired features of the images in progressively smaller layers, during which pooling layers are used for each convolutional layer to down-sample the output. The latter one is used for up-sampling the output of the previous layer. To reconstruct output images, the last convolutional layer concatenated the previous layer with the same size as the input images [34]. During the "up" component, the detailed global features of the convolutional layer and the contextual information of the previous layer are combined together, which are then fed to the next convolutional layers. U-net has attained promising results in biomedical image segmentation [32,35], building extraction [36,37] and road network extraction from aerial imagery [33,38].

U-net combines low-resolution information (providing object-based identification) and high-resolution information (providing segmentation-based identification), so it is ideal for extracting perceptual features of HRRS images for generating perceptual hash sequences. In this study, we use an improved U-net model with a novel training sample generation method to extract the robust edge feature of HRRS images.

## 3. Proposed Scheme

This paper presents a deep-learning-based perceptual hash scheme, named perceptual hash based on modified U-net for the authentication of HRRS images. The architecture of our proposed scheme consists two parts, as shown in Figure 2: (1) Training stage, which builds training samples and trains the model to extract the robust edge feature of HRRS images. (2) Authentication stage, which generates perceptual hash values based on the pre-trained U-net model and implements integrity authentication for HRRS images.

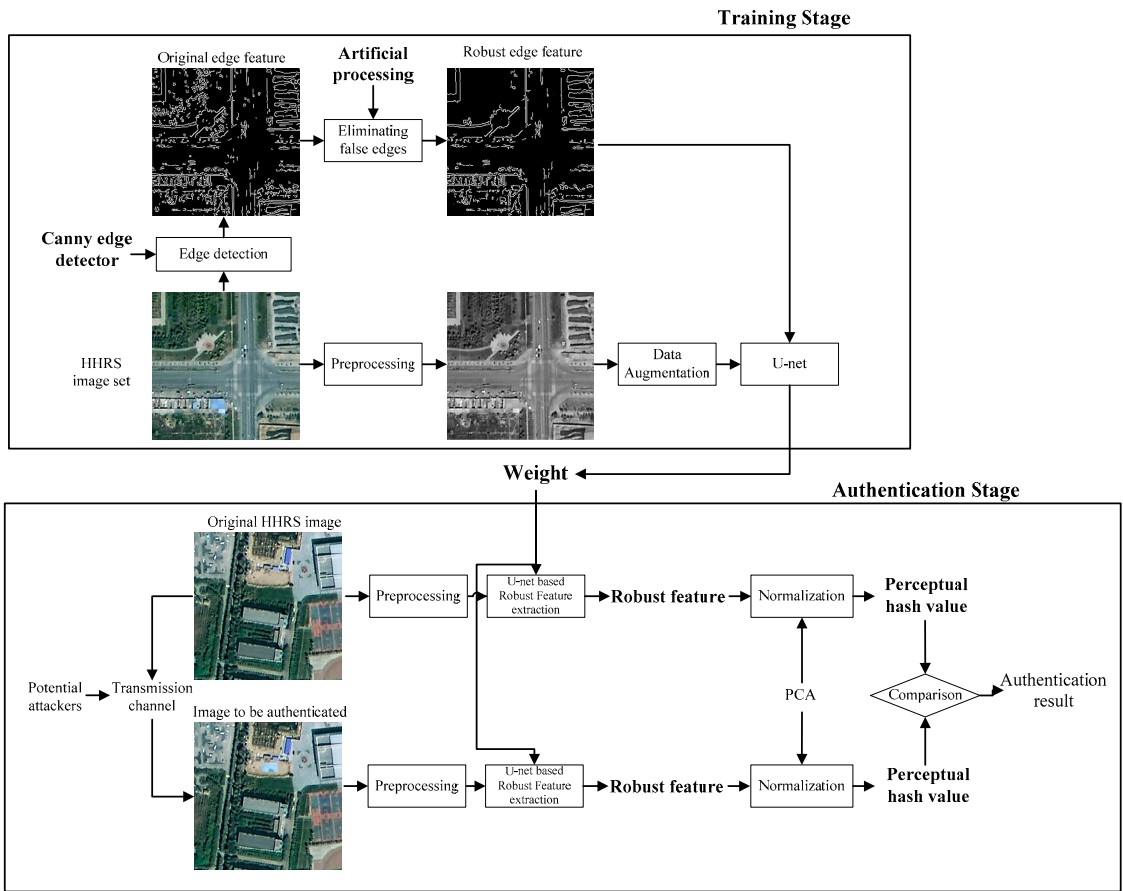

**Figure 2.** Framework of the proposed scheme.

### 3.1. Training Stage

In the training stage, the initial edge features of the training samples are detected by Canny operator firstly. To obtain robust edge features, the edge features are processed manually, for example, deleting false edges, adding partial edge features, etc. At the same time, the source HRRS images were converted into gray images. The robust edge features and the gray images constitute the training data of the modified U-net model.

#### 3.1.1. Training Sample Production Method

As the quality of the training samples has a great influence on the network model, we have to build a high-precision training sample set, that is, the effective edges of the training samples need to be accurately described. However, the hand-drawn edge images are far from accurate: Objects of the HRRS images with high resolution tend to have relatively thin edges, and they are difficult to accurately depict by hand drawing. Edges are often defined to be semantically meaningful, however, different people often have different definitions of the edges of objects, which makes the stability of the model different for different edge painters. Even the same training sample producer, its perception of

the edge of the object at different times may be biased, while the sample production takes a relatively long time.

Aiming at the above problems, we have proposed a training sample generation method that can satisfy the precise robust edge, the core idea of which is to combine artificial processing and high-precision edge extraction algorithm such as Canny operator. The training sample generation procedure is illustrated as follows.

First, edge information of the sample image is extracted by traditional and sophisticated edge extraction algorithm. At present, the gradient-based edge detection algorithms have been widely applied. Among them, the Canny operator is a widely accepted edge detection algorithm. Therefore, we used the Canny operator to extract the preliminary edge information of the samples.

Second, as the edge features extracted by Canny operator are prone to have false edge features, such as "noise characteristics", which have a certain influence on the robustness of the algorithm. We use an artificial approach to distinguish between "effective edge features" and "false features" and remove false edges from the corresponding edge images. An example of removing the false edge features for a training sample is exhibited in Figure 3, in which we can clearly see that the false edge features extracted by the Canny operator in Figure 3b are removed in Figure 3c by an artificial process.

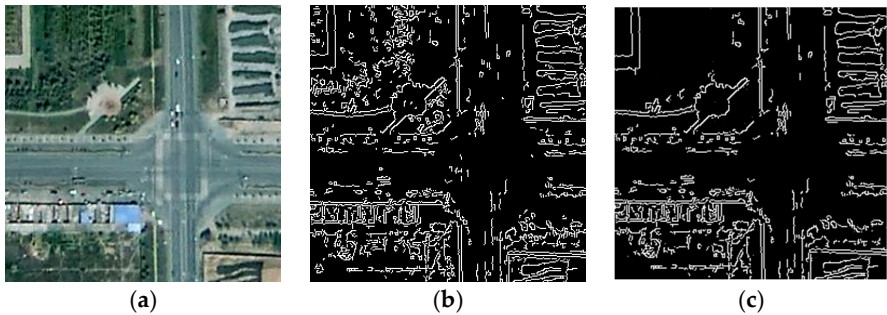

(**a**)　　　　　　　　(**b**)　　　　　　　　(**c**)

**Figure 3.** An example of removing the false edge features for training sample: (**a**) The original image, (**b**) preliminary edge features extracted by the Canny operator, (**c**) edge features after removing false edges by artificial process.

Then, due to the parameter setting of the Canny operator, some important edge information may not be detected by Canny operator. Such edges are also very important in integrity authentication, so these edge information needs to be manually added. In Figure 4b, the edge of the riverbank is not completely detected by the Canny operator, so it needs to be added manually as shown in Figure 4c.

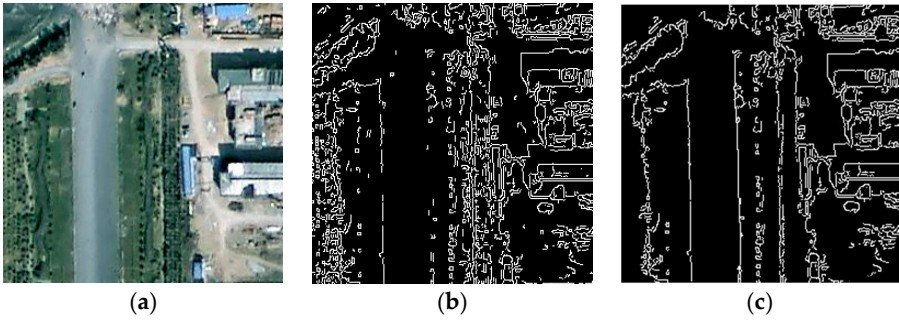

(**a**)　　　　　　　　(**b**)　　　　　　　　(**c**)

**Figure 4.** An example of adding the missing edge features for training sample processing: (**a**) The original image, (**b**) preliminary edge features extracted by the Canny operator, (**c**) edge features after adding the missing edges by manual operation.

### 3.1.2. Network Architecture

The purpose of our work is to design a new perceptual hash scheme based on deep learning, which can realize integrity authentication of HRRS images based on robust edge feature. Hence, we use a modified U-net model to extract the robust edge feature of the images. In this study, the U-net architecture is slightly different from the original U-net [33] as follows.

(1) Exponential linear unit (ELU) is applied to replace rectified linear unit (ReLU) used by the original U-net model as activation function, which improves the performance of the network, resulting in more robust and accurate edge feature extraction.
(2) Batch normalization (BN) is used in this network to accelerate the convergence. For every mini-batch, BN normalizes the inputs using the mini-batch mean/variance and de-normalizes them with a learned scaling factor and bias. It significantly reduces the internal and external variances in various attributes, and it improves the performance of the network [39–41].

Figure 5 shows the architecture of the modified U-net model for robust feature extraction, which consists of a contracting path and a symmetric expanding path. The contracting path consists of repeated application of two 3 × 3 convolutions (unpadded convolutions), each followed by one 2 × 2 max pooling operation with stride 2 for down-sampling, and one batch normalization (BN) for convergence acceleration. The expansive path consists of an up-sampling of the feature map followed by an up convolution (2 × 2) with stride 2 and two 3×3 convolutions with stride 1 in which each is also followed by batch normalization. At the same time, the correspondingly cropped feature from the contracting path will be concatenated. As exponential linear unit (ELU) is beneficial for learning representations more robust to noise [41], we leveraged ELU rather than ReLU as the primary activation function during training.

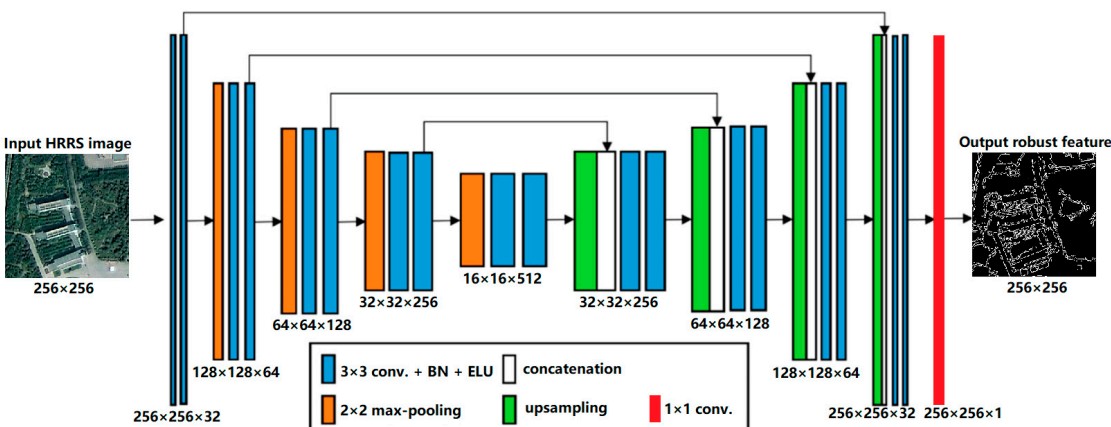

**Figure 5.** Architecture of the modified U-net network.

What is more, as the detection of edge features is essentially a binary classification problem of image pixels, that is, the pixels are either points on the edge or not, we choose binary cross-entropy as the loss function for binary classification in the model.

### 3.2. Authentication Stage

In the authentication stage, the perceptual hash value of the original HRRS image is computed and sent to the authenticator (data receiver) along with the image. If the receiver suspects that the image may have been tampered with, the authentication process is performed via the comparison between the original hash value and the reconstructed one.

### 3.2.1. Perceptual Hash Value Generation Based on Modified U-Net Model

Figure 6 shows the process framework of the perceptual hash value generation for HRRS images, which consists of two main stages. (1) A modified U-net-based robust edge feature extraction process, which extracts the robust edge feature of HRRS image. (2) A principal component analysis (PCA)-based encoder F, which reduces the dimensionality of robust features and then generates perceptual hash value PH for HRRS image.

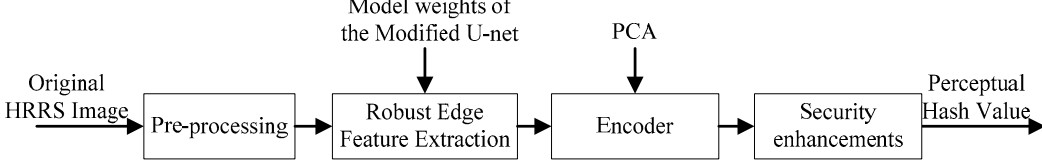

**Figure 6.** Framework of the perceptual hash value generation.

Detailed steps are listed as follows:

(1) Taking the characteristics of multi-bands of remote sensing images into account, bands of HRRS image are fused together to generate the input data for our modified U-net model. For an image with $N$ bands, the fusion model used in this paper is as follows:

$$F(x, y) = \sum_{k=1}^{N} \alpha_k R^k(x, y) \tag{1}$$

where $R^k(x,y)$ and $F(x,y)$ respectively denote the pixel values of the $K$th band $R^k$ and the fused image $F$ coordinates $(x,y)$, $\alpha_k$ denotes weighting factor. In different applications, the importance of different bands is often different, so the determination of the weighting factor $\alpha_k$ should be based on the specific application. For example, in our previous study [42], we used the information entropy of the band to determine the weighting factor for TM images. For HRRS images with three bands, we set the weighting coefficients as 0.30, 0.59, and 0.11 respectively according to the human eye's visual characteristics.

For single-band HRRS image, this band fusion operation is not required.

(2) Robust edge feature extraction.

The trained U-net models can be directly used to extract the edge feature of the pre-processed HRRS images. However, due to the limitation of the graphics memory, the fused HRRS image is normalized by interpolation algorithm (in this paper, we take bilinear interpolation as an example) into an image of $M \times M$ pixel size (in this paper, $M = 256$). Then we take the normalized image as input data for the trained U-net model, the prediction of the image resulted in a $256 \times 256$ labeled image, which is the robust edge image and is denoted as $E$. Unlike the result of the Canny operator, the labeled image $E$ is a grayscale image rather than a binary image.

(3) PCA-based encoder.

Compactness is one of the most important characteristics that perceptual hash algorithms should satisfy, which means the perceptual hash value should be as compact as possible to make it convenient to transfer, storage, and use. In the encoder phase, we did not use a machine-learning-based approach like some other studies. This is because the machine-learning-based encoding method requires a large number of training samples, and it is difficult to build a training sample set that meets the requirements due to the handcrafted robust edge images. To achieve feature compression, we use a PCA-based encoder on the feature matrix.

PCA is a common transform that is often used in the numerical analysis of matrices [43–45]. By performing PCA on the edge feature matrix, we can remove the linear correlation of the matrix element to achieve data compression and noise elimination. Before we perform PCA on the edge feature matrix, the size of the labeled image $E$ is resampled to $m_1 \times m_1$ by bilinear interpolation to achieve data compression (in this algorithm, $m_1 = 64$).

Then we serialize the resampled labeled image *E* to construct the edge feature matrix which is denoted as *ME*. As the labeled image here is a grayscale image rather than a binary image, the serialization process needs to record the decimal value of each pixel. The principal components of the matrix *ME* are then normalized to get fixed-length string, in which the principal components are binarized to a 0–1 sequence, denoted as *PString*.

To enhance the security of the authentication information itself, the string *PString* is then encrypted by using an encryption algorithm with secret *key*. In this paper, we take the 128-bit advanced encryption standard (AES-128) as an example. The encrypted string is the perceptual hash value of the original HRRS image, denoted as *PH*:

$$PH = Encrypt_{\text{AES}-128}(PString, key) \tag{2}$$

where *key* is set by the user to guarantee the security of the algorithm.

Figure 7 shows an example of perceptual hash value generation process, in which the encryption process is omitted to make the results clear. The generated perceptual hash value is a 256-bit binary sequence, and it can also be represented as hexadecimal, as shown in the Figure 7.

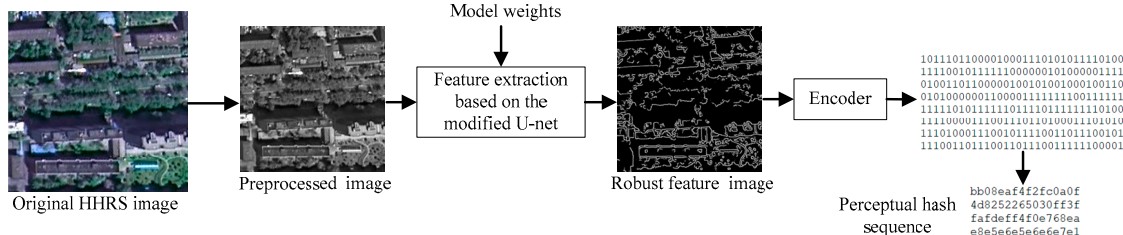

**Figure 7.** An example of the perceptual hash sequence generation (without encryption).

### 3.2.2. Perceptual Hash Process and Authentication

The authentication process is similar to the mainstream perceptual hashing algorithm. The authentication process in this paper is implemented by comparing two strings: Reconstructed perceptual hash value and the original one. A common way to determine how large the difference between the two sequences is to calculate the Hamming distance between them. However, since different algorithm parameters make the length perceptual hash values different, the number of bits with variable length cannot reflect the degree of value change very well. For example, two sequences of length 64 and 256 bits, if the same number of bits changed, it is obvious that the former is changed more. In this paper, we adopt the "normalized Hamming distance" [46] shown below:

$$Dis = \left( \sum_{i=1}^{L} \left| h_1(i) - h_2(i) \right| \right) / L \tag{3}$$

where $h_1$ and $h_2$ are perceptual hash values with *L* length. It is observed that the higher difference between the perceptual hash values', the larger the corresponding *Dis*, but not greater than 1. If *Dis* is lower than the threshold *T*, it means that the corresponding area is content-preserving. Otherwise, the content of the corresponding area has been tampered.

### 4. Experiments and Analysis

To evaluate and validate the effectiveness of the proposed deep-learning-based perceptual hash scheme for HRRS images, the datasets, experimental settings, and the experimental results are described in this section.

*4.1. Experimental Data*

To assess the efficacy of our proposed method, two sets of HRRS images are used: One is used for model training, the other is used to test the robustness and tamper sensitivity of the proposed algorithm.

Launched in August 2014, the GaoFen-2 (GF-2) satellite is the first civilian optical remote sensing satellite in China with a resolution below 1 m [47–49]. The GF-2 is equipped with two panchromatic/multispectral (PAN/MSS) cameras that can be used simultaneously, resulting in a high resolution of 0.81 m. It can obtain a 45 km width strip, which is four times that of IKNONS. Thus, GF-2 has the potential use in disaster assessments, high-accuracy navigation, mapping surveys, and other scientific research. In this paper, both of the testing set and the training set are acquired by the GaoFen-2 (GF-2) satellite with a spatial resolution of 0.8 m.

The training set consists of 50 pairs of 256 × 256 pixel HRRS images, and each pair images includes an HRRS image and a corresponding robust edge image generated based on the method in Section 3.1.1. Examples of training HRRS images are shown in Figure 8.

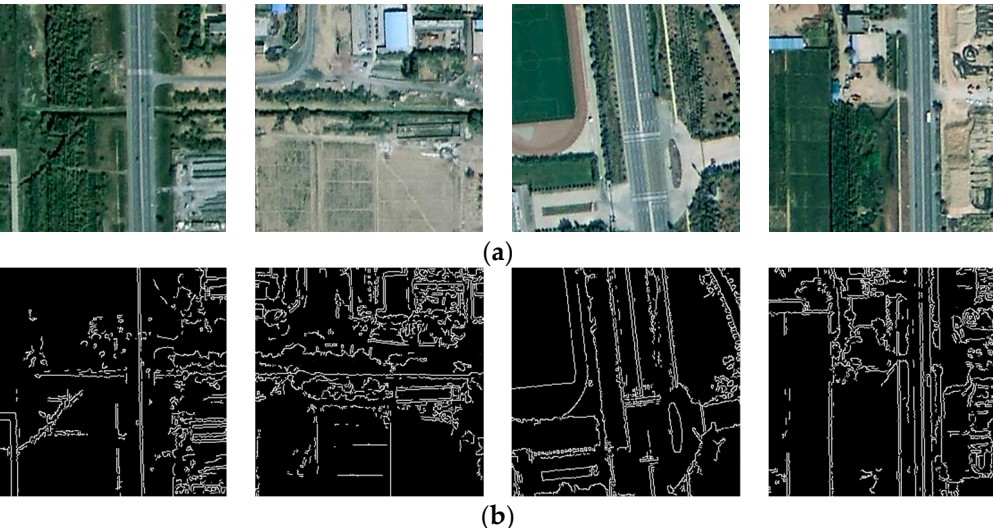

**Figure 8.** Examples of the training images: (**a**) Examples of original HRRS images, (**b**) corresponding images of robust edge feature.

In this paper, since the perceptual hash algorithm requires enough data to prove the robustness of the algorithm, the test sample is much more than the training sample. To build the testing set, we selected six three-band HRRS images in Tiff format denoted as image A to image F, as shown in Figure 9. Each of them is acquired by GaoFen-2 and is 4200 × 5000 pixels in size. Due to the limitation of the graphics memory, we divide each of them into 16 × 19 images, and then use a bilinear interpolation algorithm to resize them to 256 × 256 pixel. Eventually, we produced a testing set of 1824 HRRS images.

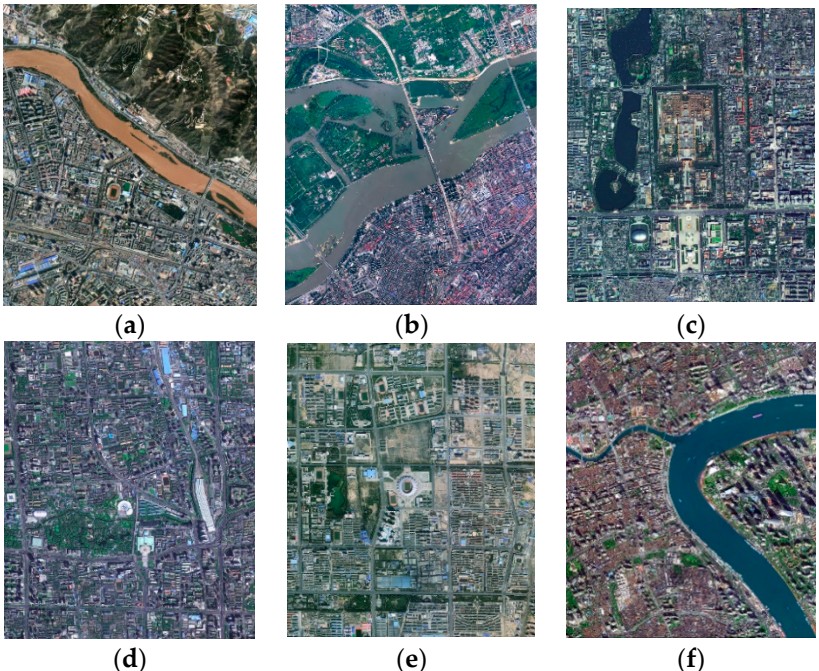

**Figure 9.** The testing HRRS images used in the experiment: (**a**) Image A (4200 × 5000 pixels), (**b**) image B (4200 × 5000 pixels), (**c**) image C (4200 × 5000 pixels), (**d**) image D (4200 × 5000 pixels), (**e**) image E (4200 × 5000 pixels), (**f**) image F (4200 × 5000 pixels).

*4.2. Implementation*

Our experiments were conducted on Ubuntu 16.04, equipped with NVIDIA GTX 1070 GPU, 8 GB of RAM. Our model was trained and tested using the Keras library with TensorFlow as the backend. The parameters of the network were set as follows: The total training epoch was 100, learning rate was $10^{-4}$, activation function was exponential linear unit (elu), and optimizer algorithm for gradient descent was Adam [50]. We trained the modified U-net network on the training data sets and tested the perceptual hash algorithm the testing data set. The training time was about 4 h.

Since the extraction of robust features is the key step for perceptual hashing, we present some examples of the results of robust edge extraction of the modified U-net model, as shown in Figure 10. Although the traditional Canny operator can extract the edge features of the image, it also extracts many false edges (as shown in Figure 10b), which will have a negative impact on the robustness of the algorithm. Of course, one can also adjust the parameters of Canny operator to avoid detecting these false edges, but this will miss many effective edge features. It can be seen from Figure 10c that the robust feature extraction algorithm of our modified U-net can largely ignore the influence of noise features such as false edges and effectively maintain effective edge features.

Moreover, the result of the Canny operator detection is a binary image, that is, each pixel can only be an edge point or not. The robust feature extraction result of the proposed algorithm is actually a grayscale image, so that each pixel is a valid edge point according to a certain weight, which is beneficial to the robustness of the algorithm.

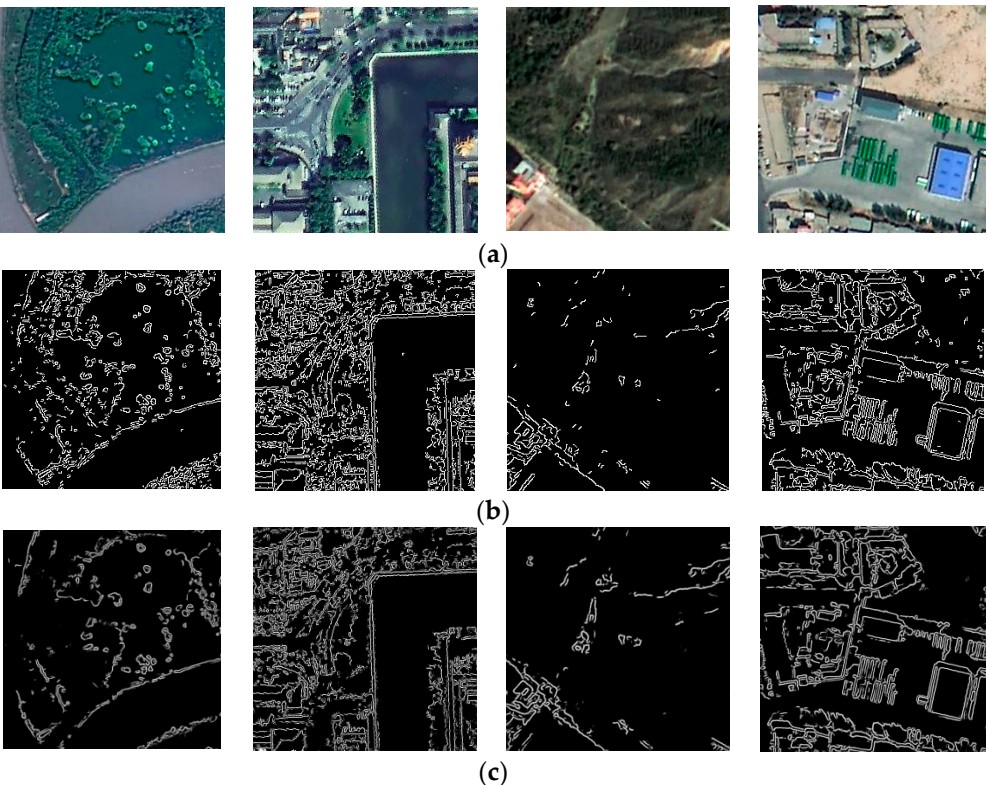

**Figure 10.** Comparison between the robust features extracted by U-net and the features extracted by Canny operator: (**a**) Examples of the original HRRS images, (**b**) edge detected by Canny operator, (**c**) robust edge detected by the modified U-net model.

### 4.3. Robust Experiments and Analysis

Robustness is the most significant difference between the perceptual hash algorithm and cryptographic hash algorithm. For HRRS images, robustness means that the perceptual hash values of the images with the same content should be the same or very similar. The important goal of our algorithm is to improve the robustness of the algorithm while maintaining the tamper sensitivity. In this study, we use the modified U-net model to learn the previously constructed training samples based on deep learning theory. If the corrected HRRS images (including geometrically corrected and radiation corrected) have undergone content-preserving operations, such as digital watermark embedding or format conversion, the perceptual hash values of the original image $I_o$ and the processed one $I_p$ should satisfy:

$$Dis(PH(I_o), PH(I_p)) < T \tag{4}$$

where $T$ is the pre-determined threshold.

Firstly, we take two content-preserving operations, watermark embedding and data format conversion, as examples to test the robustness of our algorithm. For data format conversion operations, we take the conversion between TIFF and BMP as an example. For digital watermark embedding, the least significant bit (LSB) is taken as an example. Here, the pre-determined threshold is set as 0. For the six selected HRRS images, we take the percentage of the divided images whose perceptual hash value has not changed to describe the robustness of the algorithm. As shown in Table 1, all of the perceptual hash values of the divided images have not been changed. In other words, our algorithm can keep well robust to data format conversion of TIFF to BMP and LSB watermark embedding. In contrast, cryptographic hash algorithms treat the above manipulation as illegal operations and cannot perform well for the authentication of HRRS images.

**Table 1.** The results of the robustness test1 ($T_h$ is set to 0).

| Manipulation | Format Conversion (TIFF to BMP) | Digital Watermarking (LSB) |
|:---:|:---:|:---:|
| Image A | 100% | 100% |
| Image B | 100% | 100% |
| Image C | 100% | 100% |
| Image D | 100% | 100% |
| Image E | 100% | 100% |
| Image F | 100% | 100% |

Next, we test the robustness of the algorithm to data compression operations, which is compared with the algorithm in [10]. Here, the lossless compression (PNG compression), 95% JPEG compression, and 99% JPEG compression are taken as examples. In [10], the threshold $T$ is set to 0.20. Here, to make it easier to compare the results, both of the thresholds of this algorithm and the algorithm in [10] are set to 0.10. The proportion of images that the *Dis* is greater than $T$ are shown in Table 2.

**Table 2.** The results of the robustness test (Both of the $T_h$ are set to 0.10).

| Manipulation | JPEG Compression (99%) | | JPEG Compression (95%) | | Lossless Compression (PNG Compressing) | |
|:---:|:---:|:---:|:---:|:---:|:---:|:---:|
| | Algorithm in [10] | This Algorithm | Algorithm in [10] | This Algorithm | Algorithm in [10] | This Algorithm |
| Image A | 78.6% | 89.5% | 72.4% | 83.1% | 100% | 100% |
| Image B | 80.3% | 89.1% | 75.3% | 82.6% | 100% | 100% |
| Image C | 79.6% | 90.8% | 74.9% | 82.2% | 100% | 100% |
| Image D | 81.3% | 91.4% | 74.0% | 83.2% | 100% | 100% |
| Image E | 80.6% | 92.4% | 78.3% | 87.2% | 100% | 100% |
| Image F | 81.9% | 90.1% | 79.3% | 83.9% | 100% | 100% |

It can be seen from Table 2 that the robustness of the proposed algorithm to data compression operations is improved compared to the algorithm in [10], with an average increase of 10%. The robustness of the algorithm can be adjusted by changing the algorithm parameters such as $T$. It should be pointed out that robustness only makes sense if the algorithm can effectively detect the malicious tampering of HRRS images. If robustness is excessively emphasized, sensitivity may be directly affected and some tampering of HRRS image may be impossible to identify, which affects the authentication accuracy of HRRS image. Since the ability of conventional image perceptual hash algorithms to detect tampering is insufficient to meet the authentication requirements of HRRS images, the robustness of the proposed algorithm is not compared with them.

In addition, the robustness of the proposed algorithm depends largely on the production of the training sample. If the training sample deletes too many edges, the robustness will be enhanced, but it will affect the tamper sensitivity of the algorithm.

It should be noted that the authentication object of our algorithm is the corrected (including geometric correction and radiation correction) HRRS image, the rotation and scaling robustness are out of our consideration.

*4.4. Sensitivity to Tampering Experiments and Analysis*

Sensitivity to tampering means that the algorithm can effectively detect the tampering of the HRRS images. In this study, if the HRRS image has been tampered, the perceptual hash values of the original image $I_o$ and the processed one $I_p$ should satisfy:

$$Dis(PH(I_o), PH(I_p)) > T \tag{5}$$

To test the sensitivity to tampering, we take several kinds of tampering operations, including changing object, appending object, removing object and optional local subtle tampering, to test the sensitivity to tampering. For each kind of tampering operation, we construct a dataset with 50 tampering instances. Some examples of each kind of tampered HRRS images are shown in Figures 11–14.

The more popular image perceptual hash algorithm mainly includes the methods based on discrete cosine transform (DCT) [4,51–53], the method based on discrete wavelet transform (DWT) [9,54,55] and the method based on singular value decomposition (SVD) [5,56]. In the related research specifically for HRRS images, the algorithms in [10] are the best algorithms currently published. We compare the above algorithm with our proposed algorithm, and the comparison results are shown in Table 3. Here, we take the percentage of successfully detected tampering instances to describe the sensitivity to tampering. The threshold of the normalized Hamming distance for the method based on wavelet transform is set to 0.02, the threshold for the method based on DCT is set to 0.02, the threshold for the method based on SVD is set to 0.01, and the threshold for the algorithms in [10] and our proposed algorithm are set to 0.10.

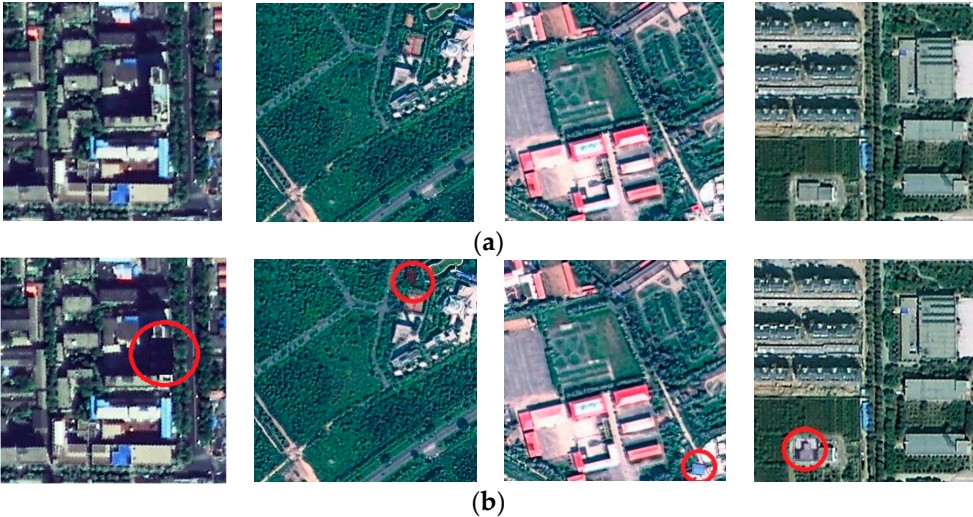

**Figure 11.** Tampering examples of changing object: (**a**) Original HRRS images, (**b**) corresponding tampered images.

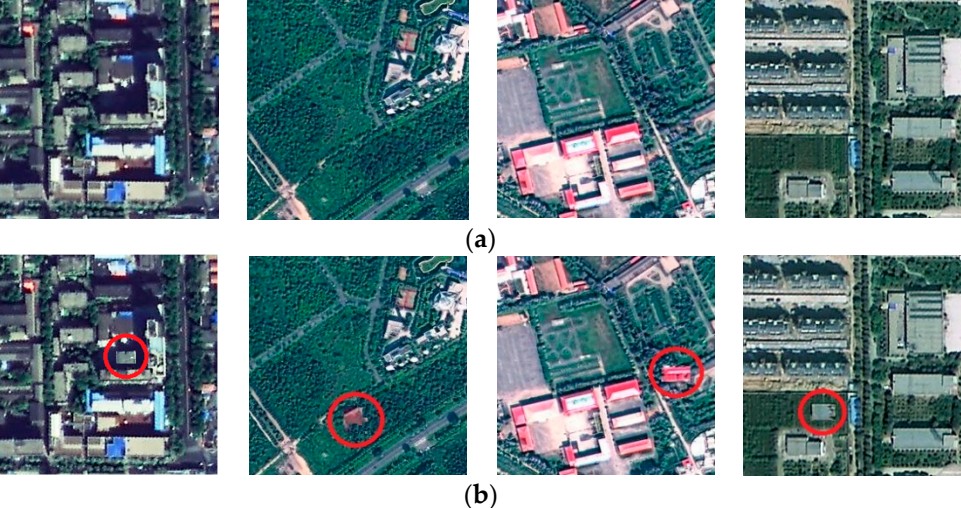

**Figure 12.** Tampering examples of appending object: (**a**) Original HRRS images, (**b**) corresponding tampered images.

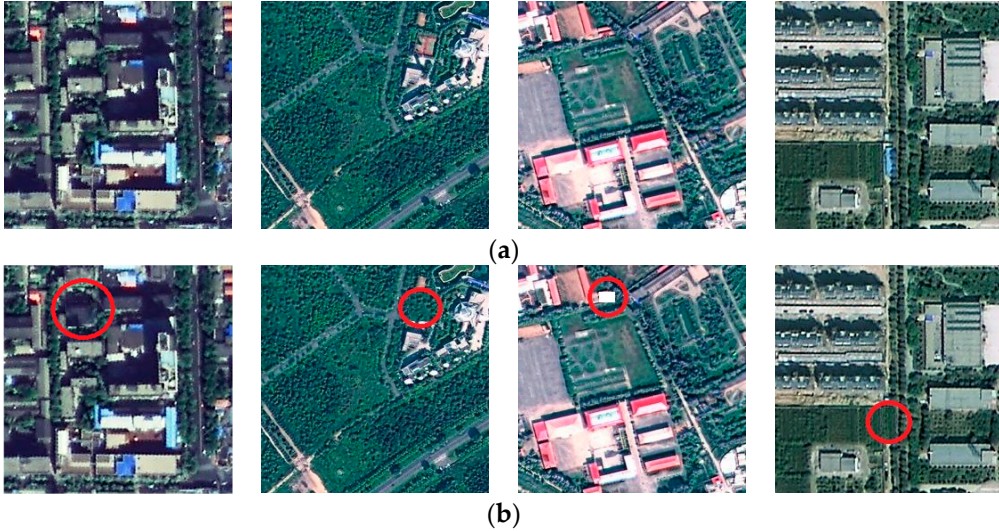

**Figure 13.** Tampering examples of removing object: (**a**) Original HRRS images, (**b**) corresponding tampered images.

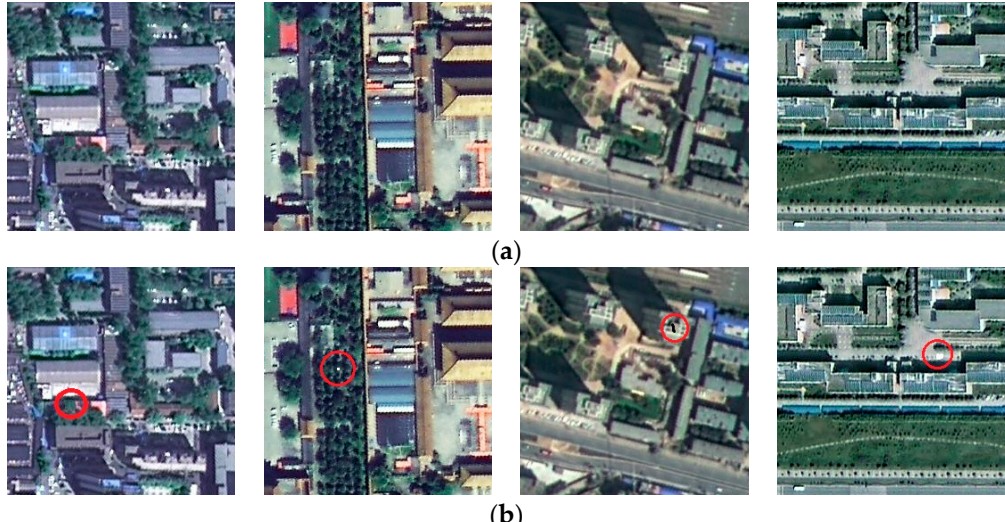

**Figure 14.** Examples of local subtle tampering: (**a**) Original images, (**b**) corresponding tampered images.

**Table 3.** The comparison of sensitivity to tampering.

| Tampering Test | Algorithm Based on DCT | Algorithm Based on DWT | Algorithm Based on SVD | Algorithm in [10] | This Algorithm |
|---|---|---|---|---|---|
| Removing the object | 40% | 44% | 42% | 100% | 100% |
| Appending the object | 52% | 56% | 56% | 100% | 100% |
| Changing the object | 42% | 36% | 38% | 100% | 98% |
| Local subtle tampering | 8% | 8% | 6% | 96% | 94% |

It can be seen from Table 3 that our algorithm has similar tampering sensitivity to the algorithm in [10] and is capable of detecting subtle tampering of HRRS images, meeting the high-accuracy requirements of HRRS images. In contrast, the tampering sensitivity of traditional algorithms is far from achieving the integrity authentication requirements of HRRS images. In practical applications, if a higher tamper sensitivity is required, a smaller threshold $T$ can be set so that more subtle tampering can be detected, which may affect the robustness of the algorithm.

*4.5. Analysis of Algorithm Security*

The security of perceptual hash algorithm mainly refers to the unidirectionality, that is, no valid content information of the image can be obtained from the perceptual hash value without the key. For our algorithm, the security relies on the security of the encryption algorithm, and we adopt the advanced encryption standard (AES-128) algorithm in this paper. As the security of the AES-128 algorithm has been widely studied and recognized, our algorithm has enough security. Moreover, if there is a higher requirement for the security of the perceptual hash value, a stronger encryption algorithm such as aes-256 can be used.

## 5. Discussion

The authentication of HRRS image is essentially concerned with the integrity of the effective content of the image, rather than the changes in the data carrier itself. Therefore, the traditional authentication methods, such as cryptography-based methods and digital-watermark-based methods, do not adequately address the problem of authentication for HRRS images. Although perceptual hash algorithms can achieve perceptual-content-based authentication for conventional image data, the existing research results still have large deficiencies for the authentication of HRRS images, especially in terms of tamper sensitivity and robustness.

Tamper sensitivity and robustness are the two most important attributes of perceptual hashing, but the two are often contradictory. The key to solving this contradiction lies in how to extract the robust features of the image. Based on deep learning theory, we introduce the classic network model U-net to improve the robustness of perceptual hash algorithm while maintaining the tamper sensitivity. Another important contribution of our algorithm is that we propose a training sample generation method combining Canny operator and artificial processing to meet the high accuracy requirement of training samples for authentication.

Through the above experiments in Sections 4.2–4.4, it can be seen that compared with conventional perceptual hash algorithms and the algorithm in [10], the advantage of our algorithm is that the algorithm has better robustness and maintains favorable tamper sensitivity. Moreover, the training of the modified U-net model requires only 50 pairs of training samples, which basically meets the expectation.

Of course, although the algorithm has better tamper detection capability than most image perceptual hash algorithms, it cannot accurately detect all of the malicious tamperings. In the experiments in Section 4.4, a few malicious tampering instances escaped the detection of our algorithm. In fact, the tamper detection capability of our algorithm can be adjusted by changing the algorithm parameters or threshold. For example, we can set the threshold $T$ lower to improve the sensitivity of the algorithm to detect more subtle malicious tampering, which will have a certain negative impact on the robustness of the algorithm, or, in the phase of PCA-based feature compression, we can extract more principal components of the feature matrix to generate longer perceptual hash values, but it may affect the compactness of the algorithm. In summary, if the algorithm is over-emphasizing the ability to detect subtle tamperings, other aspects of the algorithm would be affected.

At present, although our work obtained a relatively good result, it is still possible to be improved in future work. (1) HRRS images tend to be relatively large in size, and often cannot be authenticated as a whole like conventional images but are segmented into small images for authentication. Thus, how to ensure the continuity of information between sub-images is a question that needs further study. (2) Tampering positioning allows the perceptual hash algorithm to not only authenticate the content integrity of the HRRS image, but also determine where the HRRS image has been tampered with. However, the tampering positioning accuracy of the existing perceptual hash algorithms is not ideal. Therefore, higher precision tampering positioning method still needs further research.

## 6. Conclusions

In this paper, we have proposed a deep-learning-based perceptual hash scheme for the authentication of HRRS images. Different from traditional perceptual hash algorithms with classic

feature extraction method, we use a modified U-net model for robust feature extraction. What is more, to meet the high precision requirements of training samples, a novel training sample generation method combining artificial processing and Canny operator was proposed to extract the robust edge features. The proposed scheme consists of two main stages: Training stage and authentication stage. In the training stage, training samples were built, and the model of the modified U-net was trained. In the authentication stage, the perceptual hash values of HRRS images were generated and used to implement integrity authentication. The experiments show that this algorithm has strong tampering sensitivity and has some improvement on the robustness to content-preserving operations.

In future work, we intend to study the applicability of the perceptual hash algorithm to different resolution images, the information continuity problem caused by block authentication of HRRS images and how to locate the tampered area intelligently based on deep learning.

**Author Contributions:** K.D. conceived the idea and worked together with Z.Y. to design the scheme; Y.W., and Y.L. assisted with the study design and the experiments. All authors reviewed the manuscript.

**Funding:** This study is patricianly supported by the grants from: (a) the National Key R&D Program of China (Grant No. 2016YFF0204006); (b) the National Natural Science Foundation of China (Grant No. 41801303); (c) the Jiangsu Province Science and Technology Support Program (Grant No. BK20170116); (d) the Scientific Research Hatch Fund of Jinling Institute of Technology (Grant Nos. jit-fhxm-201604, jit-b-201520); and (e) the Qing Lan Project.

**Conflicts of Interest:** The authors declare no conflict of interest.

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
