# Peer review of "An Improved Perceptual Hash Algorithm Based on U-Net for the Authentication of High-Resolution Remote Sensing Image"

_applsci, doi:10.3390/app9152972_

Reviewer 1 Report

In this paper, the authors tackle the issue of detecting tempered high resolution remote sensing images. To do so they propose a Deep Learning model based on PCA and a modifier U-Net architecture to generate perceptual hash values

The paper is relatively well writen and contains all adequate references.

However, I have serious doubts about the efficiency of the proposed method: U-Net is a black box neural networks and a non-deterministic algorithm, and PCA is a compression technique used for visualization and that does not handle non-linearity or high dimensionality. As such, I don't see how the proposed architecture could detect subtle alterations in HRRS images that would basically be absorbed by these compression methods.

Even the robust edge feature extraction presented in Figure 2 shows that this is not perfect enough to detect small or subtle alterations. As such, it seems to me that the proposed methods falls short of the expectations given in the introduction.

It seems to me that up to page 13, all the authors do is showing that U-nets are better than Canny at edge detection, which is a very well established fact.

On the other hand, the real interesting results are in page 13 to 16 and should be better used as they show what I suppose is the most important in this paper.

Minor remarks:

Other flaws include the lack of description of the used U-Net architecture. I suggest replacing lines 280 to 296 (page 8) with a Figure.

Figure 1 should be modified like Figures 10 to 13 to better see where the alterations are.

Author Response

Dear Reviewer, 

Thank you very much for your constructive comments.

Responds to your comments: 

In this paper, the authors tackle the issue of detecting tempered high resolution remote sensing images. To do so they propose a Deep Learning model based on PCA and a modifier U-Net architecture to generate perceptual hash values

The paper is relatively well written and contains all adequate references.

1. However, I have serious doubts about the efficiency of the proposed method: U-Net is a black box neural networks and a non-deterministic algorithm, and PCA is a compression technique used for visualization and that does not handle non-linearity or high dimensionality. As such, I don't see how the proposed architecture could detect subtle alterations in HRRS images that would basically be absorbed by these compression methods.

Reply:

For the perceptual hash algorithm, it not only needs to detect malicious tampering effectively, but also needs to be robust to the operation of not changing the image content. That is, the perceptual hash algorithm should only detect the operation of changing the perceptual content of the image, and not be very sensitive to pixel-level changes. What's more the perceptual hash value generated by the algorithm should be as compact as possible. A short perceptual hash value is convenient for storage, transfer, and use. In this way, a short perceptual hash value cannot detect all changes of the image, but only to detect the malicious operation that change the perceptual content of the image.

In this paper, our algorithm generates a perceptual hash value based on the robust edge features of HRRS images, and use a PCA-based approach to reduce the extracted robust edge features to generate a compact perceptual hash value. For the operation of changing the effective content of the HRRS image, our algorithm needs to detect as much as possible, but if the alteration is too subtle to affect the use of the HRRS image, no recognition is needed, otherwise the robustness of the algorithm will be affected. In theory, as long as the robust edge features of the image do not change significantly, we do not think that the HRRS image has been maliciously tampered; but if the robust edge feature changes significantly, the perceptual hash sequence will also change accordingly, and malicious tampering can also be detected.  It is not necessary to detect all edge changes of the image, but to detect the tampering that affects the perceptual content of the HRRS image. In the extreme case, if the subtle alterations shrinks to a point, it would will no longer have perceptual meaning and should not be detected. Here, we determine whether the change is significant by setting the threshold Th. In other words, our algorithm only needs to detect the subtle alterations that affect the content of the HRRS image.

Of course, although the algorithm has better tamper detection capability than other image perceptual hash algorithms, it cannot accurately detect all of the malicious tampering. In the experiments of the paper, a few malicious tampering instances escaped the detection of our algorithm.

In fact, the tamper detection capability of the algorithm can be adjusted by changing the algorithm parameters and thresholds. For example, in the phase of compression PCA-based compression for robust features, we can extract more principal components of the feature matrix to generate longer perceptual hash values, but it may affect the compactness of the algorithm; or, in the tamper detection phase, we set a lower threshold, but it may affect the robustness of the algorithm. In summary, if the algorithm is over-emphasized to detect subtle alterations, other aspects of the algorithm will be affected.

In addition, we have added relevant notes to the paper, such as section 4.4.

2. Even the robust edge feature extraction presented in Figure 2 shows that this is not perfect enough to detect small or subtle alterations. As such, it seems to me that the proposed methods falls short of the expectations given in the introduction.

Reply :

Tamper sensitivity and robustness are the two most important attributes of perceptual hash, but the two are often contradictory. The perceptual hash algorithm for the authentication of HRRS image must be sensitive enough to subtle changes or tampering of the image. However, this sensitivity is essentially contradictory to the robustness of the perceptual hash. Moreover, HRRS images generally have the characteristics of massive data. If the perceptual hash algorithm over-emphasizes the sensitivity of tampering, it will inevitably increase the complexity of the algorithm and reduce the compactness of the algorithm.

As far as our algorithm is concerned, it not only needs to be robust, but also has sufficient sensitivity to malicious tampering, that is, it has to take into account the contradictory attributes of algorithm robustness and tamper sensitivity. If we simply emphasize the ability of the algorithm to detect subtle tampering, then the meaning of "perception" will be lost. For example, as long as the edge information of the image changes by a few pixels, we think that the image has been tampered, then the algorithm is no longer robust.

We all know that robustness is the biggest difference between perceptual hash and cryptographic hash. If our algorithm has a large lack of robustness because of its tamper sensitivity, the algorithm will lose its meaning. It can be seen from the experimental results that compared with the algorithm of reference [10], the advantage of our algorithm is that the algorithm has better robustness.

The problems solved in this paper mainly include: limited training samples, existing research based on deep learning mainly for image retrieval, how to interpret the detected tampering, and the accuracy of manually labelled edges is too low.

In the introduction of this paper, the core problem that this paper hopes to solve is: enhance the robustness of the algorithm while maintaining the tamper sensitivity of the algorithm; there are no ready-made and sufficient training samples for the training of the network model. In the experiments of this paper, we can see that the robustness of the algorithm is improved compared with the existing algorithms, and the tamper sensitivity is similar to the existing algorithms. Moreover, the training of this model requires only 50 pairs of training samples, which basically meets expectations.

What’s more, we have added relevant notes to the paper, such as section 2.1 and 4.3.

3. It seems to me that up to page 13, all the authors do is showing that U-nets are better than Canny at edge detection, which is a very well established fact.

On the other hand, the real interesting results are in page 13 to 16 and should be better used as they show what I suppose is the most important in this paper.

Reply :

We conducted a more in-depth analysis of the experimental part to better illustrate the features and advantages of the algorithm.

4. Other flaws include the lack of description of the used U-Net architecture. I suggest replacing lines 280 to 296 (page 8) with a Figure.

Reply :

We have added a Figure denoted as Figure 5 to illustrate the architecture of the modified of U-net.

5. Figure 1 should be modified like Figures 10 to 13 to better see where the alterations are.

Reply :

The tempered areas in figure 1 have been circled.

Thank you for your time and consideration.

Yours sincerely,

Kaimeng Ding

Reviewer 2 Report

The paper “A perceptual hash algorithm based on U-net for the authentication of high-resolution remote sensing image” is an interesting research topic which secure high resolution remote sensing images from tampering. However, there are many important issues which lessens the value of this paper and must be tackled to improve the paper content.

1-       The paper title indicates that U-net and deep learning is playing a major role in this research, but from the content in paper [10] and this paper I see many similarities except the added U-net where its role is not clear. I suggest to change the title to a more general one such as “ An Improved Perceptual Hashing algorithm Based on U-net for the Authentication of High Resolution Remote Sensing Image”

In addition, the authors should emphasize more on U-net role and use. The following remarks may help in the remediation of this issue. Do not forget to reduce plagiarism too.

2-       In the abstract mention previous work and the improvement in this work and add final results as percentage of improvements.

3-       In figure 1 show where are the tempering by putting circles around tempered objects in each image.

4-       In the introduction lines 67 to 124 can be reduced and authors can get directly to the point of why they have used U-net and perceptual hash scheme in concise way.

5-       In lines 113 which problems to solve are they the problem of using deep learning please mention that.

6-       Support the findings in lines 183 to 185 with references.

7-       From line 259 to line 264 I do not find any new idea except artificial process which is not explained!

8-       Line 277 replace word "rubust" with "robust"

9-       Line 287 to 296 can be easily understood if a graph is add.

10-   Rename subsection 3.2 to perceptual hash process and authentication.

11-   Line 314 to 321 shows the preprocessing step by combining different bands of the HRRS based on a predefined weights. How these weights are selected what if you have 17 bands with different spatial resolutions such as Worldview 3 or Gaofen-2 with four multispectral bands and a panchromatic.

12-    Lines 323 to 328 is not clear enough. Are the authors using the interpolation to reduce the size of the image or to reduce the spatial resolution or both?

13-   Is PCA used on the original image or the preprocessed and already reduced image size? In that case why PCA is needed?

14-   It is better to move the data subsection to data and method section and not in the experimental results. In addition authors should add more information about the specification of the RS sensors used in GaoFen-2 or other HRRS images.

15-   Line 390 replace the word "select" with "selected".

16-   In Lines 391 to 393 the authors indicated that images A to F with each has a size of 4200X5000 pixels and each is divided into 16x19 images with size 256x256 for each sub-image using bilinear interpolation. First is all the bands in GaoFen-2 have similar spatial resolutions? If not how the authors handled this issue? Second the authors mean that to get 16X19 sub-images with size 256x256 bilinear interpolation is used this means that values of new pixels are estimated from neighbor pixels. This means that the original images are altered with more estimated pixels will this affect the authentication process?  How do you ensure the continuity of information between sub-images after interpolation?

Author Response

Dear Reviewer, 

Thank you very much for your constructive comments.

Responds to your comments:

The paper “A perceptual hash algorithm based on U-net for the authentication of high-resolution remote sensing image” is an interesting research topic which secure high resolution remote sensing images from tampering. However, there are many important issues which lessens the value of this paper and must be tackled to improve the paper content.

1. The paper title indicates that U-net and deep learning is playing a major role in this research, but from the content in paper [10] and this paper I see many similarities except the added U-net where its role is not clear. I suggest to change the title to a more general one such as “An Improved Perceptual Hashing algorithm Based on U-net for the Authentication of High Resolution Remote Sensing Image”

In addition, the authors should emphasize more on U-net role and use. The following remarks may help in the remediation of this issue. Do not forget to reduce plagiarism too.

Reply :

We have changed the title to “An improved perceptual hash algorithm based on U-net for the authentication of high-resolution remote sensing image”.

The goal of our algorithm is to improve the robustness of the algorithm while maintaining the tamper sensitivity. Therefore, the primary purpose of introducing deep learning theory and U-net model is to enhance the robustness of the algorithm. Specifically, the algorithm of this paper creates a training sample set by combining manual processing with the canny operator. These samples remove most of the false features, which are important reasons for the robustness of the algorithm. Then, we make the U-net model to learn how to extract robust features based on the training samples. Of course, we made some changes to the original U-net model, such as replacing relu with elu as the activation function.

What’s more, we have added relevant notes to the paper, such as section 4.3.

2. In the abstract mention previous work and the improvement in this work and add final results as percentage of improvements.

Reply :

We have added the previous work and the improvements in the abstract, and used percentages to illustrate the improvement of the experimental results in this paper.

3. In figure 1 show where are the tempering by putting circles around tempered objects in each image.

Reply :

The tempered areas in figure 1 have been circled.

4. In the introduction lines 67 to 124 can be reduced and authors can get directly to the point of why they have used U-net and perceptual hash scheme in concise way.

Reply :

We have streamlined the above content in the paper.

5. In lines 113 which problems to solve are they the problem of using deep learning please mention that.

Reply :

We have modified the corresponding statement.

The problems solved in this paper mainly include: limited training samples, existing research based on deep learning mainly for image retrieval, how to interpret the detected tampering, and the accuracy of manually labelled edges is too low.

6. Support the findings in lines 183 to 185 with references.

Reply :

We added the corresponding references and changed the statements to some extent.

7. From line 259 to line 264 I do not find any new idea except artificial process which is not explained!

Reply :

Existing sample labeling algorithms often use manual drawing or labeling, but we believe that these methods can not meet the accuracy requirements of the proposed algorithm. Therefore, we adopt the method of “manual elimination of edges”, which is specifically designed for the algorithm of this paper. From this perspective, we think that there is some innovation.

8. Line 277 replace word "rubust" with "robust"

Reply :

We are sorry for this mistake and have corrected this error.

9. Line 287 to 296 can be easily understood if a graph is add.

Reply :

We have added a Figure denoted as Figure 5 to illustrate the architecture of the modified of U-net.

10. Rename subsection 3.2 to perceptual hash process and authentication.

Reply :

We have renamed subsection 3.2 to perceptual hash process and authentication.

11. Line 314 to 321 shows the preprocessing step by combining different bands of the HRRS based on a predefined weights. How these weights are selected what if you have 17 bands with different spatial resolutions such as Worldview 3 or Gaofen-2 with four multispectral bands and a panchromatic.

Reply :

In our previous research, we have conducted related research on multi-spectral images, see the paper "A Novel Perceptual Hash Algorithm for Multispectral Image Authentication" (Algorithms 2018, 11, 6.). In that paper, the weighting coefficients of the bands when they are fused are mainly determined by the entropy of the corresponding bands. Therefore, if the algorithm of this paper is applied to a large number of bands, you can refer to this method that entropy determines the weighting factor.

We also added some corresponding statements to the paper.

12. Lines 323 to 328 is not clear enough. Are the authors using the interpolation to reduce the size of the image or to reduce the spatial resolution or both?

Reply :

We use an interpolation algorithm to reduce the size of the image. In fact, for most perceptual hashing algorithms, in order to get uniform perceptual hash values, the images are generally preprocessed. In the process of preprocessing, an interpolation algorithm is often used to unify the image size. For the algorithm in this paper, we unify the image size not only to get a uniform sequence, but also because of the input of the U-net model.

We have modified some of the statements in the paper to make the statement clearer.

13. Is PCA used on the original image or the preprocessed and already reduced image size? In that case why PCA is needed?

Reply :

In this paper, PCA is used on the robust edge feature matrix and extracts the principal components of the feature matrix as the perceptual features of the original HRRS image.

The role of PCA is mainly two aspects: On the one hand, to generate a compact hash value, PCA implements the dimensionality reduction of the robust feature matrix, and extracts the principal components of the feature matrix to represent the effective content of the original HRRS image; on the other hand, PCA The process of dimensionality reduction can effectively remove part of the noise, and this denoising can further enhance the robustness of the algorithm. In fact, existing perceptual hashing algorithms often use PCA to remove the noise of the features. In summary, the role of PCA is to enhance algorithm compactness and robustness.

14. It is better to move the data subsection to data and method section and not in the experimental results. In addition authors should add more information about the specification of the RS sensors used in GaoFen-2 or other HRRS images.

Reply :

We have added more information about the GaoFen-2 satellite. Since this paper uses manual processing combined with the Canny operator to generate training samples, which is considered to be a relatively novel method in the hash field, we take the section of experimental data processing as part of the experiment. Of course, if the reviewer insists that it is necessary to separate the data section, we can also make changes.

15. Line 390 replace the word "select" with "selected".

Reply :

We are sorry for this mistake and have corrected this error.

16. In Lines 391 to 393 the authors indicated that images A to F with each has a size of 4200X5000 pixels and each is divided into 16x19 images with size 256x256 for each sub-image using bilinear interpolation. First is all the bands in GaoFen-2 have similar spatial resolutions? If not how the authors handled this issue? Second the authors mean that to get 16X19 sub-images with size 256x256 bilinear interpolation is used this means that values of new pixels are estimated from neighbor pixels. This means that the original images are altered with more estimated pixels will this affect the authentication process?  How do you ensure the continuity of information between sub-images after interpolation?

Reply :

We divide the images A to F into a number of small images in order to obtain enough test examples to test the robustness of the proposed algorithm. Robustness is important for performance evaluation of perceptual hashing algorithms, while robust testing requires sufficient examples to support it. On the other hand, the robustness of the perceptual hash also determines that it does not implement integrity authentication on a pixel-by-pixel basis, but rather implements image authentication based on perceptual content.

However, we have not conducted in-depth research on the authentication of high-resolution images with different resolutions. We intend to study this issue in the next step. Moreover, remote sensing images tend to be relatively large in size, and often cannot be authenticated as a whole image, but are segmented into small images for authentication, while conventional images tend to be authenticatied as a whole. Thus, the reviewer proposed "how to ensure the continuity of information between sub-images." is a question that needs further study. We list the issues in the conclusion of the paper as the focus of our next study.

Thank you for your time and consideration.

Yours sincerely,

Kaimeng Ding

Round  2

Reviewer 2 Report

The authors have answered most of the reviewer's concerns and questions. In addition, the authors have made the necessary modifications to make their paper more appealing to reader and scientifically correct.

Author Response

Dear Reviewer,

Thank you for your time and constructive comments.

Yours sincerely,

Kaimeng Ding